## Perspective

health and disease and epidemiology/ biogeography/mathematical modelling

fairer, Greater Manchester, life expectancy, COVID, inequality, build back fairer

**Author for correspondence:**
Michael Marmot
e-mail: m.marmot@ucl.ac.uk

# Building back fairer in Greater Manchester and the country

Michael Marmot

Institute of Health Equity, Department of Epidemiology and Public Health, University College London, London, UK

MM, 0000-0002-2431-6419

A summary of our analyses in Greater Manchester (GM), and the northwest (NW) region, might be: the NW is like England as a whole only more so. The life expectancy drop in England in 2020 was 1.2 years in men and 0.9 years in women—shocking, but not as high as in the NW. COVID-19 mortality rates were high in England; 25% higher in the NW. Inequalities in mortality are high in England; bigger in the NW. The title, *Build Back Fairer*, is a deliberate echo of the Build Back Better mantra, showing that the levels of social, environmental and economic inequality in society are damaging health and well-being. As the UK emerges from the pandemic, it would be a tragic mistake to re-establish the status quo that existed pre-pandemic—a status quo marked in England, over the decade from 2010, by a stagnation of health improvement that was more marked than in any rich country other than Iceland and the USA; by widening health inequalities; and by a fall in life expectancy in the most deprived 10% of areas outside London. That stagnation, those social and regional inequalities, and deterioration in health for the most deprived people are markers of a society that is not meeting the needs of its members.

## 1. Introduction

*The Guardian*'s front page on 30 June 2021 had two stories. The big one was headlined: England 2 Germany 0. The lesser one had the headline: 'Jaw-dropping fall in life expectancy in poor areas'. I confess to the cliché 'jaw-dropping'—my summary of a headline from our report published that day, *Build Back Fairer in Greater Manchester* [1]. In 2020, life expectancy in the northwest (NW) region of England fell by 1.6 years in men and 1.2 years in women. To put this in perspective, for 100 years or so, up to 2010, life expectancy in England had *improved* about 1 year every 4 years. In that context, the 2020 declines are a striking reversal.

A summary of our analyses in Greater Manchester (GM), and the NW region of which GM is a part, might be: the NW is like England as a whole only more so. The life expectancy drop in England as a whole in 2020 was 1.2 years in men and 0.9 years in women—also shocking, but not as high as in the NW. COVID-19 mortality rates were high in England; but 25% higher in the NW. Inequalities in mortality are high in England as a whole; bigger in the NW.

We called our report, *Build Back Fairer,* as a deliberate echo of the Build Back Better mantra. The justification for this title was, as we said in the report, that the levels of social, environmental and economic inequality in society are damaging health and well-being. As the UK emerges from the pandemic, it would be a tragic mistake to attempt to re-establish the status quo that existed pre-pandemic. As documented in our February 2020 report, *Health Equity in England: the Marmot Review 10 Years On,* that status quo was marked in England, over the decade since 2010, by a stagnation of health improvement that was more marked than in any rich country other than Iceland and the USA; by widening health inequalities; and by a fall in life expectancy in the most deprived 10% of areas outside London [2]. That stagnation, those social and regional inequalities, and deterioration in health for the most deprived people are markers of a society that is not functioning to meet the needs of its members.

## 2. Build back fairer and levelling up

When we, the UCL Institute of Health Equity, published *Build Back Fairer in Greater Manchester,* we proposed it as an evidence-based agenda for the Prime Minister's ambition to level up. Based on our understanding of the reasons for the health disadvantage in more deprived areas, a review of the evidence and an assessment of how to build back fairer, we made practical recommendations of what needed to be done. In Boris Johnson's speech on levelling up on 11 July 2021, he said: 'even before covid hit, it is an outrage that a man in Glasgow or Blackpool has an average of ten years less on this planet than someone growing up in Hart in Hampshire or in Rutland. Why do the people of Rutland live to such prodigious ages? Who knows—but they do'. Perhaps he had not had time to read our reports or the decades of research on which they were built.

The Prime Minister went on to say of London: 'the inequalities were so acute that when I became mayor in 2008 you could travel from Westminster to Canning Town on the jubilee line and lose a year of life expectancy with every stop and yet at the end of my time as mayor that was no longer true—life expectancy had increased across the capital—but the gains had been greatest among the poorest groups and that is what I mean by levelling up'. This, of course, raises the question: if no one knows why people in poorer cities, or poorer parts of cities, have shorter life expectancy, how did Boris Johnson, as mayor of London, manage to level up. The apparent contradiction is resolved in the figure. He did not level up. He was right about the Jubilee Line metaphor; not about having done anything about it. Evidence is more revealing than fantasy. The figure comes from *Health Equity in England: the Marmot Review 10 Years On* [2]. It is for women; the pattern for men is similar. It shows the clear gradient in life expectancy, the more deprived the area, the shorter the life expectancy—illustrated by stops on the jubilee line. It also shows that the gradient did not change from 2010–2012, the period shortly after Johnson became mayor, to 2016–2018, the period shortly after he left. In London, there was, it is true, improvement in life expectancy at all levels of deprivation, but no change in the gradient.

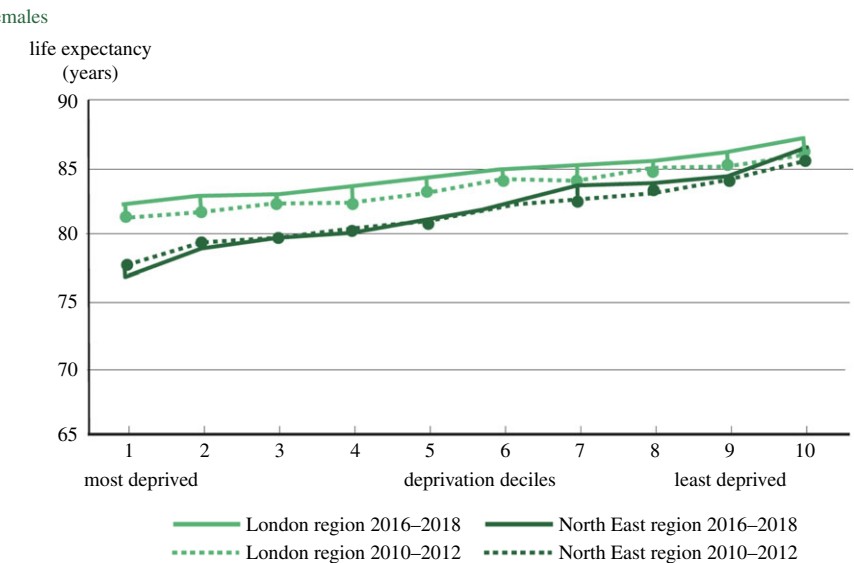

**Figure 2.7. Life expectancy at birth by sex and deprivation deciles in London and the North East regions, 2010–12 and 2016-18**

The regional contrast is highly relevant both to our GM report and levelling up. The figure shows data from the northeast region; the NW region is similar. The social gradient in life expectancy is steeper in the

north of the country than it is in London; health inequalities are bigger. In general, for people in the least deprived decile, there is little regional variation in life expectancy. If you are rich, it does not much matter where you live. The greater the deprivation, the greater the health disadvantage of living in the northeast or NW, compared to London. Further, in London, as part of there being little change in the gradient, life expectancy for the poorest people improved. By contrast for virtually every region outside London, life expectancy declined for the poorest 10% and the gradient became steeper.

# 3. Health inequalities before and during the pandemic

To give context to *Build Back Fairer in Greater Manchester*, I need to go back to 2010, when we published *Fair Society, Healthy Lives: the Marmot Review* [3]. Based on a review of the social determinants of health inequalities, we developed six domains of recommendations to improve health and reduce inequalities:

— give every child the best start in life;
— education and lifelong learning;
— employment and working conditions;
— a minimum income for healthy living;
— healthy and sustainable environments in which to live and work;
— taking a social determinants approach to prevention.

Note that I am discussing health, not healthcare. Economists and international organizations commonly speak about funding for health, when what they mean is funding for healthcare. My view of the evidence is that funding for 'health' should include funding for the six domains listed above. The opening line of my book, *The Health Gap*, asked: why treat people and send them back to the conditions that made them sick? [4]. Universal health coverage is a goal endorsed by most countries and international organizations, such as WHO and the World Bank. Healthcare systems are important for health, when people get sick but, in general, it is not lack of healthcare that leads to people getting sick in the first place; it is the social determinants of health. The increase in funding for the NHS in the decade from 2010 at just over 1% a year in real terms was markedly lower than the trend in funding in the preceding decade. This relatively miserly settlement made work in the NHS more difficult, and was indicative of a government intolerant of the public sector. By itself, it was not the reason for a worsening health picture.

Our report in February 2020, just before the pandemic crashed upon us, was a look at what had happened in the decade since 2010. As summarized above, the news was not good. A life expectancy improvement of 1 year every 4 years, which had gone on for a century, slowed dramatically in 2010–2011; health inequalities increased; and the prospects of survival for people living in the most deprived 10% of areas deteriorated. As shown in the figure, this particularly affected people in the north of England.

In GM, the link between deprivation and life expectancy was clear. Rank the 10 cities of GM from most deprived, Manchester, to least deprived, Trafford, and there was a close correlation with life expectancy, in the years 2017–2019.

Trends in the economy and society will have contributed to these trends—stagnation in wages for those without a university education, the rise of the gig economy, continued effects of industrial decline. That said, an obvious question to ask, less easy to answer, is whether the policies of the government elected in 2010 played a part in this miserable health picture. Certainly, the government was clear about rolling back the state and imposing austerity. Public expenditure was steadily reduced: from 42% of GDP in 2010 to 35% in 2019. This reduction was achieved in a regressive way. In our *10 Years On* report, we quoted analyses from the Institute of Fiscal Studies (IFS): from 2010 spending per person by local authorities declined—the greater the deprivation the steeper the reduction in spending. Thus, in the least deprived 20% of areas, spending per person went down by 16%; in the most deprived quintile, by 32%. In our 2010 *Marmot Review*, we coined the phrase proportionate universalism. We advocated universalist policies with effort proportionate to need. What we actually saw in local government expenditure, post-2010, was the opposite: effort inversely proportionate to need. The greater the deprivation the greater the need, the greater the need the greater was the *reduction* in spending per person. This regressive approach to local government spending had a bigger impact in the NW than in the more prosperous southeast of the country.

There were adverse trends in most of the six domains of recommendations that formed the backbone of our 2010 review. For example, child poverty, defined as living in a household at less than 60% of the median income, went from 27% (after housing costs) in 2010 to 30% by 2019.

Then came the pandemic. It was predictable that the pandemic would expose and amplify underlying inequalities in society. Mortality from COVID-19 in the NW was 25% higher than the English average.

Within GM, there was a social gradient in mortality from COVID-19—the more deprived the area, the higher the mortality, that looked similar to the pattern for all-cause mortality. For all-cause mortality, the most deprived decile had 2.1 times the mortality of the least deprived; for COVID-19, the ratio was 2.3. It means that the causes of inequalities in COVID-19 are similar to the causes of inequalities in health, more generally. The slightly bigger inequalities in COVID-19 associated with deprivation are probably related to employment in front-line occupations and living in overcrowded households.

## 4. Build back fairer in England and in GM

England's management of the pandemic was poor [5]. One way of comparing countries, given international differences in mode of completion of death certificates, is to examine excess mortality: extra deaths in 2020 compared to prediction, based on the previous 5 years. In the first half of 2020, excess mortality in the UK was the highest of any rich country, even higher than the USA that, under Trump, handled the pandemic very badly. In working out what 'building back fairer' should look like, it is important to ask if there is a link between England's poor state of health pre-pandemic and poor handling of the pandemic. My speculation is that the link could work at four levels: quality of governance and political culture; the level of social and economic inequalities; disinvestment in public services and poor health status in England pre-pandemic that would increase risk during the pandemic.

Addressing all four of these influences at the national level is vital. An important question underlying *Build Back Fairer in Greater Manchester* is what can happen at city and regional levels. There is a negative and a positive reason for seeking action at the local level. The negative reason is that if central government is careless, or worse, about health inequalities, action has to come from somewhere. As we documented, the actions of central government in England after 2010 were a likely contributor to stalling of life expectancy increase and of worsening health inequalities.

The positive reason for seeking action at local level is that there is a great deal of enthusiasm and commitment for actions that will improve the lives of residents—action that is, in effect, action on the social determinants of health. For example, in the wake of the 2010 *Marmot Review,* Coventry declared itself a 'Marmot City', implementing our recommendations at city level.

The work in GM extended this approach to a city region and was much influenced by the pandemic. We proposed a Build Back Fairer Framework, shown below:

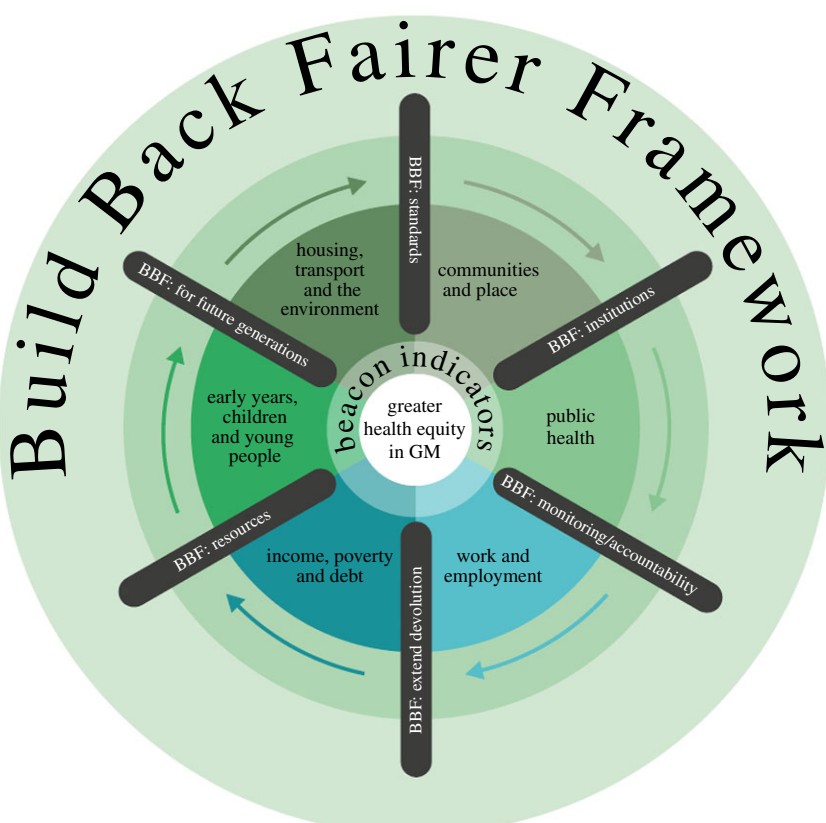

With slightly different titles, the contents of our recommendations are similar to the six domains of recommendations in the 2010 review. The six spokes are crucial steps to take in achieving progress on these dimensions: build back fairer for future generations; build back fairer resources; build back fairer standards (covering quality of employment, environment and housing, transport and clean air); build back fairer institutions (anchor institutions are a good example—considering the wider community and environmental impacts of hospitals and industry); build back fairer monitoring and accountability; build back fairer through greater local power and control. The beacon indicators, 24 in number, are a way of monitoring progress.

# 5. Evidence-based hopefulness

A common complaint takes the form of: We have known about health inequalities in the UK, at least since the Black Report (1980); why does nothing ever happen? I do not share this cynicism for at least three reasons.

First, health for the worst off, the most socially deprived, has improved greatly since the time of the Black Report. In 1980, the average life expectancy for men in England was 70.8 and for women was 76.8. In 2020, the most deprived decile in the NW region of England had the same life expectancy as the English average, 40 years earlier. Health had improved for the most deprived.

Second, health inequalities vary in magnitude over time. We reported increases after 2010. During the decade before the 2010 government, health inequalities narrowed [6]. If the Prime Minister is serious about levelling up, he could do worse than examine what did happen in the country in the 2000s to reduce health inequalities. If health inequalities are not of fixed magnitude, and the evidence provides clear guidance as to the action needed, change is possible.

Third, and related, there is real enthusiasm at local level. Coventry as Marmot City and our GM Report are clear examples. But we, at the Institute of Health Equity, have been approached by cities and local authorities from around England, wanting help in developing strategies for action on social determinants of health to reduce health inequalities. It is an example of using scientific evidence to create fairer societies. That is why I label it 'evidence-based hopefulness'.

Data accessibility. For data supporting this Perspective, see the Build Back Fairer in Greater Manchester report https://www.instituteofhealthequity.org/resources-reports/build-back-fairer-in-greater-manchester-health-equity-and-dignified-lives.
Competing interests. I declare I have no competing interests.
Funding. The work in Greater Manchester was commissioned and funded by GM.

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
