## [Peer Review File · Royal Society Open Science]

Review History

RSOS-211454.R0 (Original submission)

Review form: Reviewer 1

Is the manuscript scientifically sound in its present form?

Yes

Are the interpretations and conclusions justified by the results?

Yes

Is the language acceptable?

Yes

Do you have any ethical concerns with this paper?

No

Have you any concerns about statistical analyses in this paper?

No

Recommendation?

Accept with minor revision (please list in comments)

Comments to the Author(s)

Excellent and timely piece

Review form: Reviewer 2

Is the manuscript scientifically sound in its present form?

Yes

Are the interpretations and conclusions justified by the results?

Yes

Is the language acceptable?

Yes

Do you have any ethical concerns with this paper?

No

Have you any concerns about statistical analyses in this paper?

No

Recommendation?

Accept as is

Comments to the Author(s)

This is a high quality paper as would be expected from Michael Marmot. It is based on a report published in June.

My only thought is that Michael rightly points to lack of government action, in particular investment in the public sector and notably in local government as the principal factor that has contributed to stalling life expectancy and inequalities over the last ten years. But there is no mention of structural effects in the wider economy which may have had a large effect too eg stagnation in wages among those who are not college educated, insecurity at work with the gig economy, continued decline of large industry etc. Ideally these wider 'commercial' determinants would be mentioned, as covered more fully by Anne Case and Angus Deaton in the US and others, and indeed what government could do about them eg better regulation of working conditions (short of a proper economic strategy). The paper doesn't 'need' this addition for publication, but it does seem a missing element in the case made.

Review form: Reviewer 3

Is the manuscript scientifically sound in its present form?

Yes

Are the interpretations and conclusions justified by the results?

Yes

Is the language acceptable?

Yes

Do you have any ethical concerns with this paper?

No

Have you any concerns about statistical analyses in this paper?

No

Recommendation?

Accept with minor revision (please list in comments)

Comments to the Author(s)

This is an interesting commentary, building upon the author's own Build Back Fairer in Greater Manchester report. I am listing below a few relatively minor points that I think should be addressed. The emphasis placed on the distinction between health care and health expenditure is very welcome, as is the emphasis on the required actions, or areas in which health expenditure should take place. Readers may also wish to see references to specific analyses illustrating how much should be spent on those actions and what could be achieved in each are with such expenditure.

A few specific comments:

page 3, line 9, perhaps replace "fantasy" with "political statements"

page 4, line 43, austerity does not necessarily equate to "rolling back the state", that statement could be more cautiously revised

page 4, lines 44 to 47, sentence starting with "From 2010, spending per person [...]"; this is based on analyses that are not referenced in the paper, and it is not clear what categories of public expenditure are accounted for in the analysis; not all public expenditure can be linked to specific local authorities, of course

page 5, section "Build back fairer in England and in GM", this section contains several generalisations that need to be better supported by arguments and data, to prevent this paper from being perceived as a political statement itself (e.g. "management of the pandemic was poor" - is the excess mortality comparison a sufficient basis to claim that? - "actions of central government [...] were a likely contributor to stalling of life expectancy")

page 6, paragraph starting at line 46, it is not clear how this paragraph provides evidence of a life expectancy improvement for the worst off

A few typos to fix:

- page 2, line 21, draw- should read jaw-

- page 3, line 7, question mark missing

- page 4, line 26, full stop missing

Decision letter (RSOS-211454.R0)

Dear Dr Marmot

On behalf of the Editors, we are pleased to inform you that your Manuscript RSOS-211454 "Building Back Fairer in Greater Manchester and the Country" has been accepted for publication

in Royal Society Open Science subject to minor revision in accordance with the referees' reports. Please find the referees' comments along with any feedback from the Editors below my signature.

Please submit your revised manuscript and required files (see below) no later than 7 days from today's (ie 27-Sep-2021) date. Note: the ScholarOne system will 'lock' if submission of the revision is attempted 7 or more days after the deadline. If you do not think you will be able to meet this deadline please contact the editorial office immediately.

on behalf of Nick Pearce (Subject Editor)
openscience@royalsociety.org

Subject Editor Comments to Author (Professor Nick Pearce):

Comments to the Author:

The reviews are all very positive - the proposed revisions are largely minor textual amendments. We would be delighted to publish the piece as soon as a final version has been submitted.

Associate Editor Comments to Author:

Comments to the Author:

The consensus is that this timely and engaging piece is largely acceptable for publication: each reviewer has a couple of (minor) suggestions for modifications. In general, the journal asks revisions to be made within 7 days, but if an extension on this deadline is required, the author should contact the editorial office and an extension can be made.

Reviewer comments to Author:

Reviewer: 1

Comments to the Author(s)

Excellent and timely piece

Reviewer: 2

Comments to the Author(s)

This is a high quality paper as would be expected from Michael Marmot. It is based on a report published in June.

My only thought is that Michael rightly points to lack of government action, in particular investment in the public sector and notably in local government as the principal factor that has contributed to stalling life expectancy and inequalities over the last ten years. But there is no mention of structural effects in the wider economy which may have had a large effect too eg stagnation in wages among those who are not college educated, insecurity at work with the gig economy, continued decline of large industry etc. Ideally these wider 'commercial' determinants would be mentioned, as covered more fully by Anne Case and Angus Deaton in the US and others, and indeed what government could do about them eg better regulation of working conditions (short of a proper economic strategy). The paper doesn't 'need' this addition for publication, but it does seem a missing element in the case made.

Reviewer: 3

Comments to the Author(s)

This is an interesting commentary, building upon the author's own Build Back Fairer in Greater Manchester report. I am listing below a few relatively minor points that I think should be addressed. The emphasis placed on the distinction between health care and health expenditure is very welcome, as is the emphasis on the required actions, or areas in which health expenditure should take place. Readers may also wish to see references to specific analyses illustrating how much should be spent on those actions and what could be achieved in each are with such expenditure.

A few specific comments:

page 3, line 9, perhaps replace "fantasy" with "political statements"?

page 4, line 43, austerity does not necessarily equate to "rolling back the state", that statement could be more cautiously revised

page 4, lines 44 to 47, sentence starting with "From 2010, spending per person [...]"; this is based on analyses that are not referenced in the paper, and it is not clear what categories of public expenditure are accounted for in the analysis; not all public expenditure can be linked to specific local authorities, of course

page 5, section "Build back fairer in England and in GM", this section contains several generalisations that need to be better supported by arguments and data, to prevent this paper from being perceived as a political statement itself (e.g. "management of the pandemic was poor" - is the excess mortality comparison a sufficient basis to claim that? - "actions of central government [...] were a likely contributor to stalling of life expectancy")

page 6, paragraph starting at line 46, it is not clear how this paragraph provides evidence of a life expectancy improvement for the worst off

A few typos to fix:

- page 2, line 21, draw- should read jaw-

- page 3, line 7, question mark missing

- page 4, line 26, full stop missing

===PREPARING YOUR MANUSCRIPT===

===PREPARING YOUR REVISION IN SCHOLARONE===

- If you are providing image files for potential cover images, please upload these at this step, and inform the editorial office you have done so. You must hold the copyright to any image provided.
- A copy of your point-by-point response to referees and Editors. This will expedite the preparation of your proof.

- Ensure that your data access statement meets the requirements at <https://royalsociety.org/journals/authors/author-guidelines/#data>. You should ensure that you cite the dataset in your reference list. If you have deposited data etc in the Dryad repository, please only include the 'For publication' link at this stage. You should remove the 'For review' link.
- If you are requesting an article processing charge waiver, you must select the relevant waiver option (if requesting a discretionary waiver, the form should have been uploaded at Step 3 'File upload' above).
- If you have uploaded ESM files, please ensure you follow the guidance at <https://royalsociety.org/journals/authors/author-guidelines/#supplementary-material> to include a suitable title and informative caption. An example of appropriate titling and captioning may be found at https://figshare.com/articles/Table_S2_from_Is_there_a_trade-off_between_peak_performance_and_performance_breadth_across_temperatures_for_aerobic_scope_in_teleost_fishes_/3843624.

Author's Response to Decision Letter for (RSOS-211454.R0)

See Appendix A.

Decision letter (RSOS-211454.R1)

Dear Dr Marmot,

It is a pleasure to accept your manuscript entitled "Building Back Fairer in Greater Manchester and the Country" in its current form for publication in Royal Society Open Science. The comments of the reviewer(s) who reviewed your manuscript are included at the foot of this letter.

on behalf of Prof Nick Pearce (Subject Editor)
openscience@royalsociety.org

Comments to the author:
Delighted to accept the article as it is!

Appendix A

Response to Letter

Thank you to the reviewers for their comments on this paper.

Please find below some responses to the points made by reviewers, which go alongside the edited document.

- (Line 16-17) You ask for the date. I begin the sentence with “In 2020...” I’m sorry if that’s not clear. Compared to 2019, life expectancy in 2020 was lower by.... but that seems a bit clunkier than saying, “In 2020,... I think the last sentence also clears up any ambiguity.
- (Line 18) You suggest 2.5 years every decade. Why is that preferable to 1 year every 4 years?
- (Line 46) I have changed the punctuation as you suggested. But what I had was the punctuation in the transcript of his speech - goodness knows why No 10 should make that error - That is the punctuation in the transcript of his speech.
- (Line 58) Reviewer 3 suggested replacing fantasy with political statements. I am trying, gently, to suggest that what he said was true only in his imagination (he was lying). that’s not the same as a political statement - some of those are true.
- (Line 106) I do sometimes begin a sentence with “but” (against what I was taught). Do you really think it is necessary here. I also did not add “sole”, because the sentence begins with, “by itself”
- (Line 106) Reviewer 3 asked for a cost-benefit analysis of specific interventions. It would indeed be welcome. We invited economists to help us with it for several reports ...but... nothing to report. A different paper could review what might be done in this area. It is beyond the scope of this one.
- (Line 119-20) Thanks to Reviewer 2 for this suggestion.
- (Line 153) Reviewer 3 thought that there was need for justification for some of these assertions. I didn’t think it was controversial that management of the pandemic was poor. That said, there is ample documentation from Jeremy Farrar that that was the case.
- (Line 159) Responding to Reviewer 3, I made clear that this is speculation, but it based on what I summarised earlier in the piece.

More small formatting & grammatical changes are tracked in the document uploaded.